# The Effects of the Type of Exercise and Physical Activity on Eating Behavior and Body Composition in Overweight and Obese Subjects

**DOI:** 10.3390/nu12020557

**Published:** 2020-02-20

**Authors:** Eliane A. Castro, Eliana V. Carraça, Rocío Cupeiro, Bricia López-Plaza, Pedro J. Teixeira, Domingo González-Lamuño, Ana B. Peinado

**Affiliations:** 1Department of Sports Sciences and Physical Conditioning, Faculty of Education, Universidad Católica de la Santísima Concepción, Concepción 4090541, Chile; elianeaparecidacastro@gmail.com; 2LFE Research Group, Faculty of Physical Activity and Sport Sciences, Universidad Politécnica de Madrid, 28040 Madrid, Spain; rocio.cupeiro@upm.es (R.C.); anabelen.peinado@upm.es (A.B.P.); 3Interdisciplinary Centre for the Study of Human Performance (CIPER), Faculdade de Motricidade Humana, Universidade de Lisboa, 1495-688 Lisbon, Portugal; elianacarraca@gmail.com (E.V.C.); pteixeira@fmh.ulisboa.pt (P.J.T.); 4Nutrition Department, Hospital La Paz Institute for Health Research (IdiPAZ), 28046 Madrid, Spain; bricia.plaza@idipaz.es; 5Department of Pediatrics, University of Cantabria-University Hospital Marqués de Valdecilla, 39008 Santander, Spain

**Keywords:** macronutrients, energy intake, motivation to diet, motivation to exercise, weight loss program

## Abstract

The aim of this study was to examine whether a type of exercise favors better compliance with a prescribed diet, higher eating-related motivation, healthier diet composition or greater changes in body composition in overweight and obese subjects. One hundred and sixty-two (males *n* = 79), aged 18–50 years, were randomized into four intervention groups during 24 weeks: strength, endurance, combined strength + endurance and guideline-based physical activity; all in combination with a 25–30% caloric restriction diet. A food frequency questionnaire and a “3-day food and drink record” were applied pre- and post-intervention. Diet and exercise-related motivation levels were evaluated with a questionnaire developed for this study. Body composition was assessed by DXA and habitual physical activity was measured by accelerometry. Body weight, body mass index (BMI) and body fat percentage decreased and lean body mass increased after the intervention, without differences by groups. No interactions were observed between intervention groups and time; all showing a decreased in energy intake (*p* < 0.001). Carbohydrate and protein intakes increased, and fat intake decreased from pre- to post-intervention without significant interactions with intervention groups, BMI category or gender (*p* < 0.001). Diet-related motivation showed a tendency to increase from pre- to post-intervention (70.0 ± 0.5 vs 71.0 ± 0.6, *p* = 0.053), without significant interactions with intervention groups, BMI or gender. Regarding motivation for exercise, gender x time interactions were observed (*F*_(1,146)_ = 7.452, *p* = 0.007): Women increased their motivation after the intervention (pre: 17.6 ± 0.3, post: 18.2 ± 0.3), while men maintained it. These findings suggest that there are no substantial effects of exercise type on energy intake, macronutrient selection or body composition changes. After a six-month weight loss program, individuals did not reduce their motivation related to diet or exercise, especially women. Individuals who initiate a long-term exercise program do not increase their energy intake in a compensatory fashion, if diet advices are included.

## 1. Introduction

Obesity is a public health problem, given it relates numerous risk factors for cardiovascular disease and comorbidities [1]; this points out the need for further studies. It is known that healthy habits of physical activity and nutrition work together to maintain body weight at desirable levels [2]. Although diet contributes to a greater extent for short-term weight loss [3], exercise seems to be important in maintaining this loss [4]. Thus, exercise might also facilitate long-term adherence to healthy eating habits and behaviors.

Several studies have analyzed if exercise was able to modulate food intake [5,6], indicating that participation in physical activity as well as its duration and intensity, could contribute to appetite regulation [7], total calorie intake [8] and macronutrient composition of the diet [9,10], resulting in an appropriate energy balance. Other studies have also shown an association between regular physical activity and psychosocial and motivational factors related to a healthier eating behavior [11,12].

However, the type of exercise that can induce greater physiological and behavioral changes, related to eating behavior and food intake, remains unclear. It appears that long-term exercise interventions (more than 1 month) could decrease daily energy intake [5]. Regarding the intensity of effort, some authors found that more intense exercise reduced feelings of hunger during and after its practice [10]. Other authors have shown that absolute caloric intake was superior in high-intensity exercise, compared to moderate-intensity exercise [8]. In terms of the mode of exercise, most studies involved aerobic exercises and regular weight individuals [8,10], and the results are not very consistent. In addition, the literature lacks studies on the relationship between exercise and long-term diet adherence, and that consider the composition of the diet. Therefore, the present study aimed at examining if there is a type of exercise or a physical activity threshold (daily steps’ categories) that favors a better compliance with the prescribed diet, a higher eating-related motivation, and a healthier diet composition in overweight and obese subjects.

## 2. Methods

### 2.1. Participants

Data for this study were derived from the Nutrition and Physical Activity for Obesity Treatment (PRONAF). The inclusion criteria specified adult subjects, aged 18 to 50 years, who were overweight or obese (25 kg/m^2^ ≤ body mass index [BMI] ≤ 34.9 kg/m^2^), sedentary (<30 min physical activity/day and no participation in systematic strength or endurance training—moderate to high intensity more than once a week—in the year prior to the start of the study), normoglycemic, non-smoking and, if females, with regular menstrual cycles. Furthermore, participants with orthopedic limitations, records of stroke, eating disorders such as anorexia, bulimia etc., or any other diseases or medications that could have an effect on performance were also excluded from the study. From the 180 participants that comply with at least 90% of the training sessions and an 80% adherence to the diet total, 162 subjects were included in the analyses regarding the influence of type of exercise and 91 participants in the analyses concerning the influence of physical activity. All participants signed a written informed consent. The PRONAF study was approved by the Human Research Review Committee of the University Hospital La Paz (HULP) (No.NCT01116856).

### 2.2. Study Design and Intervention

The intervention lasted 22 weeks and the assessments took place one week before (baseline) and after (post) the intervention. Subjects were randomized by drawing lots to compose to one of the four intervention groups—strength training (S), endurance training (E), combined strength + endurance training (SE) and a guideline-based physical activity (PA), assuring a homogeneous distribution of age and gender among groups. The complete methodology and the diagram flow can be found in Zapico et al. [13].

#### 2.2.1. Exercise Protocols

The PA group followed general physical activity recommendations from the American College of Sports Medicine [14]. These recommendations were transmitted to the participants via a personal meeting. Exercise from this group was not supervised, like a classical intervention. Subjects in the S, E, and SE groups trained three times per week. All training sessions were carefully supervised by certified personal trainers. The exercise programs were designed according to the subject’s muscle strength and heart rate reserve (Figure 1). All exercise programs were performed in circuit. Adherence to exercise was calculated by the number of sessions completed in regard to the theoretical sessions ([sessions performed /total sessions] × 100).

#### 2.2.2. Hypocaloric Diet

All participants underwent an individualized dietary intervention based on a hypocaloric diet (25–30% reduction on the total daily energy expenditure) by expert dieticians. Macronutrient distribution was set according to the Spanish Society of Community Nutrition recommendations [15].

### 2.3. Assessments

#### 2.3.1. Physical Activity

Habitual physical activity was measured using the SenseWear Pro3 ArmbandTM (Body Media, Pittsburgh, PA, USA) and data inclusion followed the criteria published elsewhere [16]. Three daily steps’ categories considering initial data (before intervention) were established (<7500, 7500–10,000, >10,000 steps) [17]. The accelerometer recorded PA in metabolic equivalents (METs) for each 60 s time window (60 s epochs). Physical activity level was calculated using the average from the 3 valid days and expressed as MET per day. The daily energy expenditure was calculated using the Body Media propriety algorithm (Innerview Research Software Version 6.0, Body Media, Pittsburgh, PA, USA).

#### 2.3.2. Eating Behavior

All food and beverages consumed by the participants were recorded using a food frequency questionnaire and a “three-day food and drink record” [18]. Also, participants were asked to measure the amount of food consumed with household measurements (e.g., cups, teaspoons, tablespoons) or to write down the quantity written on commercial packaging. The total energy intake (TEI) or kilocalories consumed in the diet and macronutrient percentages (carbohydrate [CHO], fat and protein [PRO]) were calculated using the DIAL software (Alce Ingeniería S.L., Madrid, Spain). Diet compliance was calculated as the estimated Kcal of the diet (provided by dieticians) divided by the real Kcal intake ([estimated kcal of diet/real Kcal intake] 100), representing perfect adherence values equal to 100.

#### 2.3.3. Motivation

Motivation was evaluated with a questionnaire specifically developed for this study. This questionnaire consisted of 10 questions, rated on a 10-point Likert-type scale. Eight questions were related to diet motivation and two remaining to exercise motivation. Participants punctuated their diet-related motivation on a scale of de 0 to 10 by answering the following questions: “What is your motivation…” (1) to make changes in diet; (2) to go to dietary control appointments; (3) to reduce fat consumption (sausages, sauces, breaded, snacks, etc.); (4) to reduce the consumption of sugars (candies, chocolates, cakes, etc.); (5) to reduce the consumption of alcohol (if the participant did not consume alcohol, he should punctuate 10); (6) to increase fruit consumption; (7) to increase consumption of vegetables; (8) to increase fish consumption. Also, subjects scored their exercise-related motivation by answering the following questions: “What is your motivation…” (1) to exercise regularly; (2) to perform the physical exercise tasks recommended in the program. The total scores for diet and exercise-related motivation were calculated by summing the scores of the respective questions. Subscales of motivations to diet and exercise showed acceptable internal consistency, with Cronbach’s alphas of 0.73, and 0.84, respectively.

#### 2.3.4. Body Composition

Body composition was assessed by dual-energy X-ray absorptiometry scan (DXA) (GE Lunar Prodigy; GE Healthcare, Madison, WI, USA). Height was measured using a Seca Stadiometer (Quirumed, Valencia, Spain), which has a range of 80–200 cm. Body mass was measured using a TANITA BC-420MA balance (Bio Lógica Tecnología Médica S.L, Barcelona, Spain).

### 2.4. Statistical Procedures

Data are presented as mean ± SD. One-way analysis of variance (ANOVA) was used to analyze differences at baseline among the four intervention groups. Four-way repeated measures ANOVA were used to determine differences among the four intervention groups or among the three daily steps’ categories, BMI categories and gender by time. Bonferroni post-hoc tests were employed to locate specific differences. Partial eta-squared ($\eta_{P}^{2}$) was adopted for interactions and small, moderate and large effect corresponded to values equal or greater than 0.001, 0.059, and 0.138, respectively [19]. Effect sizes (ES) were calculated by Cohen’s *d* and interpretation based on <0.2 = trivial, ≥0.2<0.5 = small effect, ≥0.5<0.8 = moderate effect, and ≥0.8 = large effect. Pearson’s correlations separated by intervention groups were used to test associations between eating behavior, physical activity and motivations changes. All analyses were conducted in SPSS-PC v20 (IBM SPSS Statistics, Armonk, NY USA), and level of significance was set at 0.05.

## 3. Results

Participants’ baseline characteristics are described in Table 1. No differences were observed among intervention groups for any variable.

Large negative correlations between diet compliance and TEI were observed in all intervention groups (*p* < 0.001). Moderate to large correlations between motivations to diet and to exercise were also observed in all groups (*p* < 0.05), except in the strength group (Table 2).

Analysis of variance showed that all intervention groups, similarly, decreased body weight (*F*_(1,143)_ = 661.636, *p* < 0.001, ES = 0.58), BMI (*F*_(1,143)_ = 424.586, *p* < 0.001, ES = 0.96) and body fat percentage (*F*_(1,143)_ = 485.434, *p* < 0.001, ES = 0.79) and increased lean body mass (*F*_(1,143)_ = 484.113, *p* < 0.001, ES = 0.78). Men had greater body weight and lean body mass and smaller body fat percentage than women before and after the intervention (*p* < 0.001). The remaining interactions and factors were not significant (Figure 2).

Moreover, gender-time interactions (*F*_(1,137)_ = 4.783, *p* = 0.030, $\eta_{P}^{2}$ = 0.034) and BMI category-time interactions (*F*_(1,137)_ = 13.233, *p* < 0.001, $\eta_{P}^{2}$ = 0.088) for TEI were observed. Men consumed more calories in both, pre- and post-intervention than women (differences: 615.6 kcal, *p* < 0.001, ES = 0.72; 314.0 kcal, *p* = 0.002, ES = 0.66; respectively). Obese subjects showed greater TEI than overweight participants in pre- and post-intervention (differences: 685.6 kcal, *p* < 0.001, ES = 0.81; 176.9 kcal, *p* = 0.029, ES = 0.36; respectively). However, no interactions were observed between intervention groups and time and all of them decreased energy intake (*F*_(1,137)_ = 133.742, *p* < 0.001, ES = 1.06). The remaining interactions and factors were not significant. CHO and PRO intakes increased and fat intake decreased from pre- to post-intervention (CHO: *F*_(1,137)_ = 65.652, *p* < 0.001, ES = 0.81; PRO: *F*_(1,137)_ = 90.606, *p* < 0.001, ES = 1.02; fat: *F*_(1,137)_ = 116.561, *p* < 0.001, ES = 1.16) without significant interactions with intervention groups, BMI category or gender (Figure 3). The remaining interactions and factors were not significant. Diet-related motivation showed a tendency to increase from pre to post-intervention (*F*_(1,146)_ = 3.799, *p* = 0.053, ES = 0.14), without significant interactions with intervention groups, BMI category or gender. In relation to motivation to exercise, interactions between gender and time were observed (*F*_(1,146)_ = 7.452, *p* = 0.007, $\eta_{P}^{2}$ = 0.049), and only women increased their motivation after the intervention (pre: 17.6 ± 2.5, post: 18.3 ± 2.0, *p* = 0.034, ES = 0.31). There was a tendency of interactions between intervention group and time (*F*_(3,146)_ = 2.440, *p* = 0.067, $\eta_{P}^{2}$ = 0.048) and pairwise comparisons showed that only the PA group decreased motivation to exercise (*p* = 0.045, ES = 0.25). No interactions with BMI category were observed. The remaining interactions and factors were not significant (Figure 4).

When the sample was analyzed by daily steps categories (*n* = 91) similar results were observed. Anthropometric and body composition changes were in the same direction as observed for intervention groups, without differences among daily step categories: Body weight, BMI and body fat percentage decreased and lean body mass increased (*p* < 0.05). BMI category-time interactions were also observed for TEI, and obese individuals obtained greater values than overweight in the pre-intervention (difference: 765.4 kcal, *p* = 0.004, ES = 0.78). However, no interactions were observed between daily step categories and time and all of them decreased energy intake (*p* < 0.05). In addition, as observed earlier, CHO and PRO increased and fat decreased (*p* < 0.05). However, individuals who started the intervention program with more daily steps increased PRO compared to individuals who performed less than 7500 daily steps, independently of BMI category or gender (≥7500<10000: *p* = 0.011, ES = 0.69; ≥10000: *p* < 0.001, ES = 1.54). The observed tendency towards an increase in diet-related motivation in all intervention groups was, in this case, significant (*p* < 0.05) without interactions with daily steps’ categories, BMI category or gender. Gender-time interaction was also found for motivation to exercise and only women increased their motivation after the intervention (pre: 17.8 ± 2.4, post: 18.8 ± 1.8, *p* = 0.011, ES = 0.47). The remaining interactions and factors were not significant (Table 3).

## 4. Discussion

The main findings of this study were: (1) Macronutrient distribution and total caloric intake did not differ among strength, endurance, combined or guideline-based physical activity groups; (2) if diet advices are included, long-term exercise programs do not increase energy intake in a compensatory fashion; (3) individuals that started the program performing >7500 daily steps revealed greater increases in protein intake; (4) motivation to exercise increased only in women.

To the best of our knowledge, this is the first study to investigate macronutrient distribution among three different types of exercise interventions in circuit. Our results are in accordance with previous findings that have suggested there is no substantial effect of a specific exercise on macronutrient selection [5,20]. The literature is limited and there are very few trials of sufficient duration [21]. A recent meta-analysis investigating whether an increase in exercise or physical activity could alter ad-libitum daily energy intake or macronutrient composition in healthy adults, evaluated 24 randomized trials. From these 24 trials, only two evaluated different types of exercise [21]. Shaw et al. [22] compared the effects of endurance and combined training in normal weight subjects, while Bales et al. [23], compared the effects of endurance, resistance and the linear combination of both in obese individuals. As in our study, their results also pointed to a reduction in energy intake; however these studies did not find any change in the percentage contribution of each macronutrient.

Our data showed similar decreases in fat intake in all intervention groups. Many studies found consistent negative associations between fat intake and BMI or waist circumference, besides pointing to a contribution of diet composition, especially saturated fat and refined carbohydrates, in oxidative stress and inflammation [24]. Thus, decreases, observed in this study for fat intake, could contribute to a healthier life, considering that the Spanish population typically presents an excessive consumption of fats [25]. Our results also showed an increase of CHO and PRO intakes in all intervention groups. This homogeneous change among the four groups in our study could be due to the fact that all individuals received the same diet restriction and advices, which probably influenced these responses. Other factors, as exercise intensity [9], genetics [26], or social conditions [27], also could induce differences in macronutrient selection. Furthermore, conscious food choices could mask the exercise effects to consume a particular combination of macronutrients [5].

A significant decrease in the total energy intake in response to all interventions was observed in our study, demonstrating that various types of exercise and physical activity recommendations, allied to diet, can be effective in maintaining lower energy intake even after the end of the program. The energy deficits achieved in this study (about 780 kcal) can be considered clinically meaningful and relevant for weight management, since a deficit of 100 kcal and 189 kcal per day have been estimated to prevent weight regain [28]. Our results confirmed the claim that sedentary individuals who initiate a long-term exercise program do not increase their energy intake in a compensatory fashion [23], if diet advices are included.

In our study, although the sample was sedentary at the beginning of the intervention (<30 min physical activity/day), subjects who started the program with more daily steps presented an increase in protein intake, although diet recommendations were equal for all the participants. Other studies have shown that active individuals consumed more protein than inactive counterparts [10,29]. This fact could be related to the belief that active individuals need to consume more proteins than sedentary individuals [10].

Regarding motivation to diet and exercise, our findings indicated a good acceptance of the intervention program by the individuals. Although, motivation to exercise increased only in women, men did not decrease it. These findings are in line with previous studies observing differences in motivation to exercise between men and women [30]. According to this study, the reasons for exercise seem to differ between genders, with men being more motivated by competitiveness and exceeding limits, while women tend to be more motivated by aspects related to aesthetics [31]. Still, prior research has shown that better quality motivations for exercise (i.e., autonomous) can be effectively promoted in weight management interventions, contributing to sustained weight loss [32]. Weman-Josefsson et al. [33], showed that exercise autonomous (i.e., good quality) motivation was a stronger predictor of continued exercise participation for women rather than for men. Therefore, in our study, weight loss could have contributed for an increase in autonomous motivation in women. Future studies should confirm this hypothesis.

Nutritional education alone may not be sufficient for changing eating behavior [34], and variations in adopted behaviors can affect energy balance and weight loss success. In this sense, exercise seems to play an important role in inducing better energy deficit in relation to dietary restriction, through the prevention of nutritional compensatory responses [35]. Furthermore, a strong relationship between exercise and eating behavior has been observed, and individuals who reported being internally motivated to eat were significantly more likely to engage in physical activity and presented lower BMI [36,37]. In fact, studies have indicated that emotionally-driven eating was related with less engagement in physical activity [38] and higher BMI [39,40].

Although ANOVA has not showed differences among intervention groups in energy intake, macronutrient distribution or motivations to diet and to exercise, some correlation results are worth to be commented upon. Correlations between motivations to diet and to exercise were found for all groups, except for the strength group. In general, the correlations observed in the strength group showed that individuals who ate more also moved more, and those who ate less moved less. The strength group was the only one that, although not significantly, increased absolute lean mass (data not shown). Blundell et al. [41] observed that fat-free mass was positively associated with daily energy intake and hunger levels, which could explain, at least in part, some results found in this group. Guelfi and colleagues [42] found that aerobic exercise was associated with an increase in satiety, while an equivalent period of 12 weeks of resistance training was not. It is possible that resistance exercise needs more time to improve the responses related to appetite control and further research should investigate these responses in relation to chronic exercise in a longer period of time (one year or more).

Nutritional behavior plays an important role in predicting cardiovascular risk for any category of BMI, and exercise behavior is also an important predictor for the presence of normal weight or overweight [43]. Further research is required to determine the mechanisms through which exercise training may affect and improve eating behavior, besides increasing the understanding that obesity is not a simple matter of will power or that merely eating less and exercising more is the solution [44]. Desirable eating and physical activity habits can be the best allies against overweight and its consequences, without adverse effects, unlike other methods such as herbal weight-loss products [45] or medications [46]. Success in long-term weight loss and maintenance increases as more behavioral dimensions are involved in the behavior change process [47].

There are many strengths in the study, such as the randomized design; inclusion of four different training programs in the same study, combined with caloric restriction in all interventions, and inclusion of appropriate programs of strength and combined training for comparison among groups, however, some limitations can be highlighted, including; [1] non-assessment behavioral/psychological markers of eating behavior (food cravings, emotional eating, eating disinhibition, etc.); [2] non-inclusion of a ‘no treatment’ control group; but rather compared the new intervention with the previous one that has been proven to be effective (i.e., diet and exercise recommendations) and is broadly accepted from an ethical point of view and in clinical practice; [3] non-compliance with the criteria for the analysis of the accelerometry data pointed to the difficulty of the use of the equipment by the subjects, and did not allow a more complete analysis of the data to be performed.

## 5. Conclusions

In conclusion, individuals who initiate a long-term exercise program do not increase their energy intake in a compensatory fashion, if diet advices are included. Our findings suggest that there is no substantial effect of the type of exercise on macronutrient selection or energy intake, but they point out a general change on these parameters when an exercise program is followed. Significant increases in carbohydrate and protein intakes and decreases in fat and total energy intakes were reported, regardless of the type of exercise, BMI category or gender, as a result of following a weight loss program. Furthermore, in a subsample of the present study, individuals who started the intervention performing higher number of daily steps increased more their protein intake than those who performed <7500 daily steps. Lastly, there were increases in diet and exercise motivations, although motivation to exercise was incremented only in women.

## Figures and Tables

**Figure 1 nutrients-12-00557-f001:**
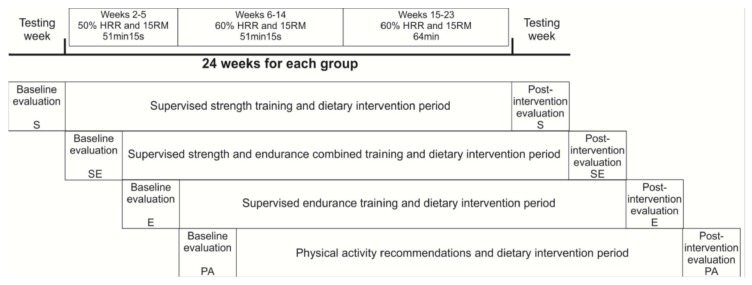
Time line of the Study. HRR: heart rate reserve; RM: repetition maximum; S: strength training group; E: endurance training group; SE: combined strength plus endurance training group; PA: physical activity recommendations group.

**Figure 2 nutrients-12-00557-f002:**
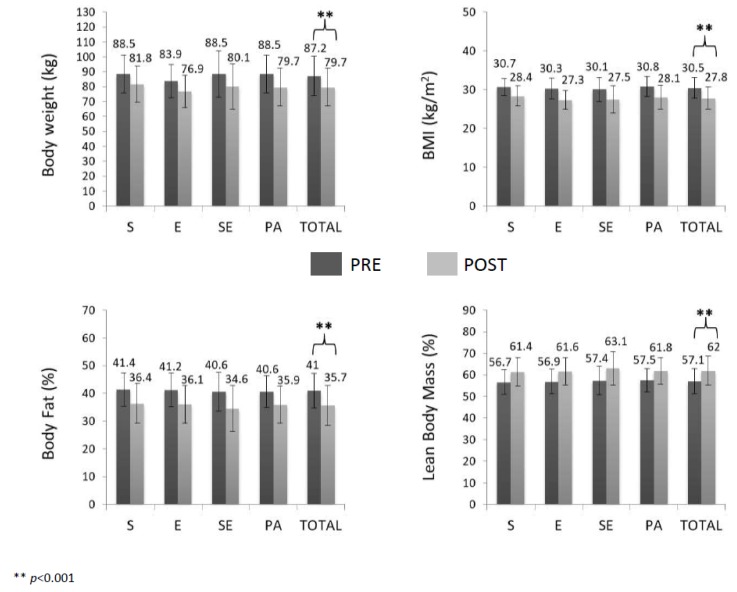
Anthropometric characteristics and body composition before and after the intervention. BMI: body mass index; S: strength training group; E: endurance training group; SE: combined strength plus endurance training group; PA: physical activity recommendations group.

**Figure 3 nutrients-12-00557-f003:**
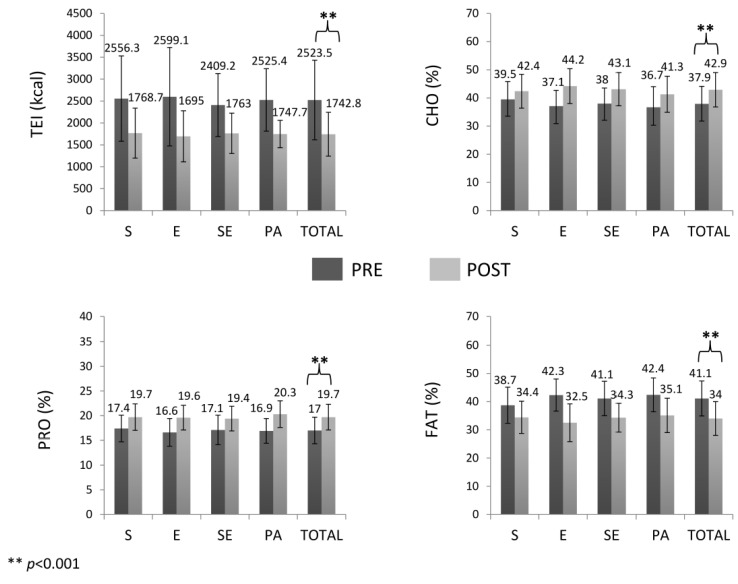
Total energy intake and macronutrient percentages before and after the intervention. TEI: total energy intake; CHO: carbohydrate intake; PRO: protein intake; FAT: fat intake; S: strength training group; E: endurance training group; SE: combined strength plus endurance training group; PA: physical activity recommendations group.

**Figure 4 nutrients-12-00557-f004:**
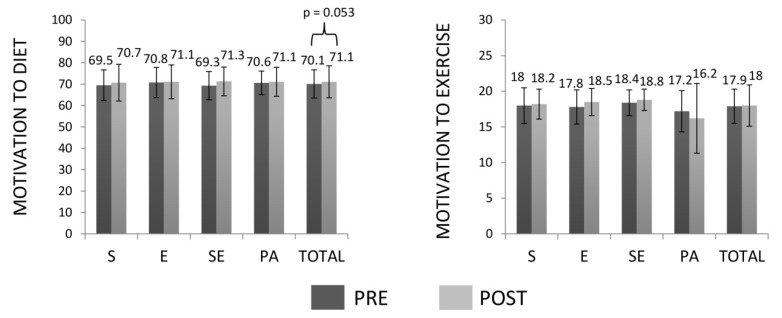
Motivation to diet and exercise before and after the intervention. S: strength training group; E: endurance training group; SE: strength and endurance training group; PA: physical activity recommendations group.

**Table 1 nutrients-12-00557-t001:** Baseline characteristics.

Variables	S (*n* = 39)	E (*n* = 48)	SE (*n* = 42)	PA (*n* = 33)	*F* (*p*)
Male (%)	46.2	41.7	50.0	60.6	0.399 ^§^
Age (yrs)	38.1 ± 8.9	37.4 ± 7.9	38.7 ± 6.6	38.4 ± 7.9	0.223 (0.88)
Body weight (kg)	88.5 ± 12.5	84.3 ± 11.2	88.8 ± 15.6	88.5 ± 12.6	1.264 (0.29)
Height (cm)	169.5 ± 9.8	167.9 ± 9.1	170.2 ± 10.3	169.5 ± 9.4	0.474 (0.70)
Body mass index (kg/m^2^)	30.7 ± 2.2	30.4 ± 2.6	30.2 ± 3.1	30.8 ± 2.6	0.399 (0.75)
Body Fat (%)	41.4 ± 6.0	41.2 ± 6.2	40.8 ± 7.0	40.6 ± 5.7	0.158 (0.92)
Lean body mass (%)	56.7 ± 5.7	56.9 ± 5.8	57.3 ± 6.6	57.5 ± 5.4	0.156 (0.93)
Total Energy Intake (kcal)	2578.4 ± 964.0	2598.74 ± 1122.8	2412.2 ± 727.0	2512.3 ± 709.2	0.335 (0.80)
PAL (METs/day)	1.4 ± 0.2	1.3 ± 0.2	1.4 ± 0.2	1.3 ± 0.1	0.785 (0.50)
Number of steps (steps/day)	11231.5 ± 3648.7	10130.1 ± 3760.1	10064.5 ± 3134.2	9409.6 ± 2189.3	1.010 (0.39)

Data are presented as mean ± SD. S: strength training group; E: endurance training group; SE: combined strength plus endurance training group; PA: physical activity recommendations group; PAL: physical activity level; METs: metabolic equivalents. ^§^ Chi-Square statistics.

**Table 2 nutrients-12-00557-t002:** Pearson’s correlations between changes in dietary variables, motivation and physical activity.

STRENGTH GROUP						AEROBIC GROUP				
	TEI	Diet Compliance	MVPA	Motivation		TEI	Diet Compliance	MVPA	Motivation	
				to Diet	to Exercise				to Diet	to Exercise
TEI	1					1				
Diet compliance	−0.93 **	1				−0.88 **	1			
MVPA	0.46 *	−0.26	1			−0.15	0.14	1		
Motivation to diet	0.10	−0.18	−0.22	1		−0.03	−0.09	0.33	1	
Motivation to exercise	0.28	−0.33 *	−0.15	0.18	1	0.08	−0.10	0.07	0.49 **	1
**COMBINED GROUP**						**PHYSICAL ACTIVITY GROUP**				
TEI	1					1				
Diet compliance	−0.88 **	1				−0.92 **	1			
MVPA	0.28	−0.14	1			−0.33	0.65 *	1		
Motivation to diet	−0.13	0.10	−0.01	1		0.29	−0.28	−0.10	1	
Motivation to exercise	0.04	−0.07	0.01	0.42 *	1	0.15	−0.08	0.03	0.43 *	1

TEI: total energy intake; MVPA: moderate-to-vigorous physical activity. * *p* < 0.05; ** *p* < 0.001.

**Table 3 nutrients-12-00557-t003:** Dietary and motivation-related variables by categories of daily steps at baseline.

	<7500	≥7500<10,000	≥10,000	Total	*F*	*p*	$\eta_{P}^{2}$
**Body Weight (kg)**					T = 419.333 TxS = 5.149 TxBMI = 1.008 TxC = 0.176 TxSxBMIxC = 1.444	<0.001 0.026 0.318 0.839 0.242	0.841 0.061 0.013 0.004 0.035
Baseline	89.1 ± 12.9	84.7 ± 10.6	85.2 ± 13.2	85.8 ± 13.2			
Post-intervention	80.6 ± 11.8	76.6 ± 10.8	77.4 ± 12.9	77.4 ± 12.9 **			

**BMI (kg/m^2^)**					T = 313.006 TxS = 0.699 TxBMI = 1.368 TxC = 0.139 TxSxBMIxC = 1.231	<0.001 0.406 0.246 0.871 0.297	0.798 0.009 0.017 0.003 0.030
Baseline	30.5 ± 2.6	29.7 ± 2.2	30.3 ± 2.7	30.1 ± 2.5			
Post-intervention	27.5 ± 2.6	26.7 ± 2	27.3 ± 2.8	27.1 ± 2.5 **			

**Body Fat (%)**					T = 299.511 TxS = 2.327 TxBMI = 0.597 TxC = 0.262 TxSxBMIxC = 0.691	<0.001 0.131 0.442 0.770 0.504	0.791 0.029 0.007 0.007 0.017
Baseline	41.3 ± 5.9	39.2 ± 6.4	40.8 ± 6.6	40.3 ± 6.4			
Post-intervention	35.7 ± 7.1	33.5 ± 7.1	34.6 ± 7.2	34.4 ± 7.1 **			

**Lean Body Mass (%)**					T = 300.088 TxS = 2.104 TxBMI = 0.512 TxC = 0.269 TxSxBMIxC = 0.615	<0.001 0.151 0.476 0.765 0.543	0.792 0.026 0.006 0.007 0.015
Baseline	56.9 ± 5.5	58.8 ± 6.1	57.3 ± 6.2	57.7 ± 6.0			
Post-intervention	62.1 ± 6.6	64.1 ± 6.7	63.1 ± 6.8	63.2 ± 6.7**			

**TEI (kcal)**					T = 41.841 TxS = 3.670 TxBMI = 6.725 TxC = 1.554 TxSxBMIxC = 2.115	<0.001 0.060 0.012 0.219 .128	0.377 0.051 0.089 0.043 0.058
Baseline	2447.9 ± 1026.8	2434.3 ± 805.2	2766.2 ± 1186.0	2592.9 ± 1044.5			
Post-intervention	1781.5 ± 448.5	1811.0 ± 563.9	1850.5 ± 651.7	1823.3 ± 580.4 **			

**CHO (%)**					T = 30.129 TxS = 1.698 TxBMI = 0.531 TxC = 0.559 TxSxBMIxC = 0.985	<0.001 0.197 0.469 0.575 0.379	0.304 0.024 0.008 0.016 0.028
Baseline	38.5 ± 7.8	36.3 ± 7.0	37.8 ± 6.2	37.5 ± 6.8			
Post-intervention	44.0 ± 6.1	43.2 ± 6.1	43.2 ± 6.3	43.4 ± 6.1 **			

**PRO (%)**					T = 36.812 TxS < 0.001 TxBMI = 1.440 TxC = 4.587 TxSxBMIxC = 0.962	<0.001 0.994 0.234 0.013 0.387	0.348 <0.001 0.020 0.117 0.027
Baseline	17.3 ± 3.5	17.6 ± 2.1	16.3 ± 2.5	16.9 ± 2.6			
Post-intervention	19.1 ± 2.5	19.2 ± 2.5	20.3 ± 2.7	19.7 ± 2.6 **			

**FAT (%)**					T = 48.246 TxS = 0.296 TxBMI = 0.716 TxC = 0.316 TxSxBMIxC = 1.674	<0.001 0.588 0.400 0.730 0.195	0.411 0.004 0.010 0.009 0.046
Baseline	40.7 ± 7.9	41.5 ± 6.9	41.1 ± 6.1	41.2 ± 6.7			
Post-intervention	33.4 ± 5.5	33.9 ± 6.0	33.4 ± 6.0	33.5 ± 5.8 **			

**MOTIVATION TO DIET**					T = 11.851 TxS = 1.838 TxBMI = 0.496 TxC = 0.704 TxSxBMIxC = 1.170	0.001 0.179 0.483 0.498 0.316	0.130 0.023 0.006 0.018 0.029
Baseline	70.0 ± 7.2	71.0 ± 6.9	70.9 ± 6.3	70.7 ± 6.6			
Post-intervention	73.1 ± 5.6	73.5 ± 6.4	72.2 ± 6.8	72.8 ± 6.4 *			

**MOTIVATION TO EXERCISE**					T = 1.422 TxS = 6.305 TxBMI = 2.325 TxC = 0.494 TxSxBMIxC = 0.609	0.237 0.014 0.131 0.612 0.546	0.018 0.074 0.029 0.012 0.015
Baseline	17.5 ± 3.6	18.4 ± 1.7	17.8 ± 2.4	18.0 ± 2.5			
Post-intervention	18.1 ± 4.5	18.2 ± 3.7	18.4 ± 2.2	18.3 ± 3.3			


TEI: total energy intake; CHO: carbohydrate intake; PRO: protein intake; FAT: fat intake; T: time (pre-post); S: gender; BMI: body mass index category; C: daily steps’ categories. * *p* < 0.05; ** *p* < 0.001.

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
