# Peer review of "The Effects of the Type of Exercise and Physical Activity on Eating Behavior and Body Composition in Overweight and Obese Subjects"

_nutrients, 2020, doi:10.3390/nu12020557_

Round 1

Reviewer 1 Report

This is a well performed study on the association between different kinds of physical exercises and better compliance with the prescribed diet, eating-related motivation, and healthier diet composition in overweight/obese individuals. The data and the topic are of interest.

The article in general fits the scope of the journal with potential good empirical contribution.

Here are some aspects that should be addressed:

Section 2.1 please specify how inclusion/exclusion criteria were assessed. Please also specify how randomization has been performed. Please clarify if (and eventually how) participants with psychiatric disease were excluded (especially those with BED). Furthermore, it is unclear why the Authors did not include the assessment of typical eating dysfunctional pattern observed in overweight/obese individuals (e.g., food craving) that could be reduced with physical activity (e.g., PMID: 27769147).

Sample size: What was the rationale for the sample size and does this produce adequate power for the measurements/analyses? Please clarify. In the data analysis, no details were reported regarding how the missing data were handled.

Section 2.3.3. Motivation questionnaire should be reported in the manuscript as supplementary material or as appendix.

Table 1. Please add sex and detailed statistics (i.e., F, chi square for sex, and p values).

Please check whether the stated conventions are really acceptable for partial eta-spared. As a common mistake, eta-squared and partial eta-spared are often reported synonymously, although values and conventions may differ depending on the statistical method and sampling (see e.g., Levine & Hullett, 2002, Human Communication Research; Pierce, Block, & Aguinis, 2004, Educational and psychological measurement; Ferguson, 2009, Professional Psychology: Research and Practice). Therefore, I would highly recommend to make sure that you do not only adopt partial eta squared from SPSS but rather calculate Cohen's d or comparable effect sizes (f, r) with clear conventions. This would allow for proper and undistorted evaluations regarding the clinical meaning of the results and enhance comparability with other studies.

Statistical Procedures: according to the recommendations for pre-post designs (PMID: 16921578), variables could be analyzed using multivariate analyses of covariance (MANCOVAs) with group as a between-subject factor, values at T1 as dependent variables, and value at T0 as a covariate. The authors should of course feel free to offer a rebuttal of this suggestion

The discussion section may be expanded emphasizing the healing benefits of physical activity compared to other practices for weight control, such as herbal weight-loss products use, which are known to be related to several negative consequences (e.g., PMID 26457296).

Author Response

RESPONSE TO REVIEWER 1 COMMENTS

Reviewer: This is a well performed study on the association between different kinds of physical exercises and better compliance with the prescribed diet, eating-related motivation, and healthier diet composition in overweight/obese individuals. The data and the topic are of interest.

The article in general fits the scope of the journal with potential good empirical contribution.

Response: Thank you for your summary and assessment. We also thank the reviewer for your constructive comments that will greatly improve our manuscript. We have tried to clarify your specific comments bellow.

Here are some aspects that should be addressed:

Point 1: Section 2.1 please specify how inclusion/exclusion criteria were assessed. Please also specify how randomization has been performed. Please clarify if (and eventually how) participants with psychiatric disease were excluded (especially those with BED). Furthermore, it is unclear why the Authors did not include the assessment of typical eating dysfunctional pattern observed in overweight/obese individuals (e.g., food craving) that could be reduced with physical activity (e.g., PMID: 27769147).

Response 1: In relation to inclusion/exclusion criteria all participants underwent a screening before starting the intervention. In this occasion, their body weight and information about lifestyle (physical activity, smoking and alcohol consumption) were recorded and only the subjects who attended the criteria were included. Furthermore, their medical histories were consulted and participants with orthopedic limitations, irregular menstrual cycles (in the case of women), records of stroke and eating disorders such as anorexia, bulimia etc., or any other diseases that could have an effect on performance were excluded from the study. Ingestion of any medication known to influence physical performance or the interpretation of results and participation in systematic strength or endurance training (moderate to high intensity more than once a week) in the year prior to the start of the study were also considered exclusion criteria. Randomization was ensured by drawing lots to allocate intervention groups. More information was included in the text in order to elucidate these issues (lines 77-81; 89).

Regarding assessment of typical eating dysfunctional pattern, this aspect was not the focus or one of the outcomes of the study. We recognize the effect of exercise on the improvement of several behavioral / psychological markers of eating behavior (food cravings, emotional eating, eating disinhibition, etc.), but in our study these were not evaluated because we choose to explore the effects of exercise in other food related variables. However, we included as a limitation of our study the fact that we have not evaluated them and, therefore, explored the effects of exercise on these variables and their interactions with others (lines 297-298).

Point 2: Sample size: What was the rationale for the sample size and does this produce adequate power for the measurements/analyses? Please clarify. In the data analysis, no details were reported regarding how the missing data were handled.

Response 2: The estimation of the sample size was calculated to detect a main effect of the treatment on the percentage of body fat with an 80% statistical power at 5% significance, assuming a 0.80 correlation between repeated measures. The initial calculated sample size per intervention group (n = 22) permits the detection of a large effect size (Cohen’s d=−0.8), as observed in a previous investigation (PMID: 19150861). The initial adjustment of sample size for dropouts carried out used a maximum of 25% of dropouts in each group (PMID: 19567540). Adjusted dropouts sample = n (1/1-R), where n = number of subjects not lost, and R = expected proportion for dropouts. With our data, the initial necessary sample size was: 88 (1 / 1–0.25) = 118 subjects or 29/30 in each intervention group (PMID: 23259716).

Regarding missing data, these were excluded from the analysis.

Point 3: Section 2.3.3. Motivation questionnaire should be reported in the manuscript as supplementary material or as appendix.

Response 3: In order to answer to another reviewer's request, questionnaire questions were included in the Motivation subsection in Methods section. Lines: 130-140.

Point 4: Table 1. Please add sex and detailed statistics (i.e., F, chi square for sex, and p values).

Response 4: Sex and detailed statistics were added in the Table 1.

Point 5: Please check whether the stated conventions are really acceptable for partial eta-spared. As a common mistake, eta-squared and partial eta-spared are often reported synonymously, although values and conventions may differ depending on the statistical method and sampling (see e.g., Levine & Hullett, 2002, Human Communication Research; Pierce, Block, & Aguinis, 2004, Educational and psychological measurement; Ferguson, 2009, Professional Psychology: Research and Practice). Therefore, I would highly recommend to make sure that you do not only adopt partial eta squared from SPSS but rather calculate Cohen's d or comparable effect sizes (f, r) with clear conventions. This would allow for proper and undistorted evaluations regarding the clinical meaning of the results and enhance comparability with other studies.

Response 5: We agree with the fact that the results may be better interpreted if expressed with Cohen’s d. We performed a complementary analysis and modified the values in the Results section, although little difference in their interpretation was observed. Lines: 167-209.

We also add the information in statistical procedures subsection in Methods section. Lines: 152-156.

Point 6: Statistical Procedures: according to the recommendations for pre-post designs (PMID: 16921578), variables could be analyzed using multivariate analyses of covariance (MANCOVAs) with group as a between-subject factor, values at T1 as dependent variables, and value at T0 as a covariate. The authors should of course feel free to offer a rebuttal of this suggestion.

Response 6: We appreciate the suggestion and agree that, in many cases, the most correct approach may be to use ANCOVA or MANCOVA for pre-post designs. However, in our study there are not differences among group at baseline for any variable (see Table 1). Moreover, we did not find a linear relationship pre-post for many of the variables, for example, for macronutrients with Pearson correlation values (r) considered low: carbohydrates (r=0.234, p=0.004), fat (r=0.147, p=0.070) and protein (r=0.263, p=0.001). The linear relationship between the dependent variable and the covariate is one of the additional assumptions for the use of ANCOVA. Finally, we consider the use of initial values as covariate to be erroneous in our study due to the fact that there is no independence between the covariate and the independent variables, i. e., we consider that the covariate is not independent of the treatment effects, which another condition recommended by some researchers for the use of ANCOVA.

Point 7: The discussion section may be expanded emphasizing the healing benefits of physical activity compared to other practices for weight control, such as herbal weight-loss products use, which are known to be related to several negative consequences (e.g., PMID 26457296).

Response 7: The suggestion was included in the text. Lines: 290-292.

Reviewer 2 Report

A very interesting study, especially due to consideration of the motivation to maintain a diet and physical activity during and after the intervention.

Author Response

Reviewer: A very interesting study, especially due to consideration of the motivation to maintain a diet and physical activity during and after the intervention.

Response: Thank you very much for your constructive feedback and assessment. We are very happy that you liked and appreciated the work.

Reviewer 3 Report

TYPE OF PHYSICAL ACTIVITY ON EATING BEHAVIOR AND BODY COMPOSITION IN OVERWEIGHT AND OBESE SUBJECTS

The implications of this work are recognized, as the number of individuals with overweight and obesity continues to grow worldwide.  As the study was 22 weeks in duration, this is a strong element, as most interventions of this nature are much shorter.  These results may guide future research study and intervention work to promote exercise and dietary intervention for weight loss and management to improve health outcomes but the lack of detail about the exercise prescriptions provided make this manuscript in need of major revision, as essential information on the exercise programs employed is not currently included.  Additionally, there are several other major and minor concerns that, if addressable, would strengthen this manuscript.

Major Concerns:

Title: The reviewer suggests inserting “The effects of” or “No effect of” at the start of the title. Physical activity and exercise are not defined as the same concept (exercise is a subset of physical activity) – as the study measures physical activity levels but also has participants engage in an exercise intervention – the title should choose the term that reflects the study findings most accurately or include both terms as appropriate

2. Abstract

There is no mention of physical activity assessment or the methods by which they were assessed.

3. Introduction

Are the authors able to expand on the duration and intensity elements of exercise programs that are associated with appetite regulation, total caloric intake, macronutrient comp… What is the direction of these relationships and what durations and intensities appear to be optimal?  This may help the reader understand why the Ex Rxs used in the present study were designed as they were. Pg 2; line 62: “Other authors have shown that absolute caloric intake was superior in high intensity exercise.” – Compared to what? Please clarify. Pg 2; line 66: Unclear what a “marker of physical activity level” is?

4. Methods

Please expand on the Ex Rx provided to the PA group beyond citing the ACSM Position Stand and include frequency, intensity, modes and duration recommendations for each type of exercise recommended (aerobic, strength & endurance, flexibility). How were the physical activity recommendations delivered to the PA group? Via educational materials, via in person meeting?  Did participants in this group track their activity levels in a measurable way? Additional detail about the S, E, and SE Ex Rx must be included, beyond that they trained 3 days per week. What aspects of physical fitness were targeted?  What modes of training were used?  Were aerobic modes weight bearing or non-weight bearing?  How many minutes were components of the session?  Were sets and reps used for strength training?  What muscle groups were targeted.  What % of 1RM did individuals train at?  How was progression for each component (aerobic, strength) determined?  I see some of this information in Figure 1 – but should be mentioned in text. Please expand on what is meant by “all exercise programs were performed in circuit.” How was adherence to the exercise program measured? What is meant by the “estimated Kcal of the diet?” Is this referring to the recommended Kcal provided by the dietitian? The Motivation Questionnaire should be included as an appendix or additional detail (example questions should be included in the manuscript). Were correlation analyses run by group (rather than controlling for group)? As the data are presented by group in Table 2, it seems as though the correlation analyses were run separately by group, rather than group being a control variable.

5. Results

Table 1 – what is Physical Activity Level? What is being represented? What are the units? Table 2 - Provide information about control variable as these are presented as partial correlations. Paragraph beginning on pg 4; line 168 – statements need p values or qualifying adjectives to show if changes are statistically significant or not.   6.  Discussion A paragraph detailing the limitations of the present study should be included.

Author Response

RESPONSE TO REVIEWER 3 COMMENTS

Reviewer: The implications of this work are recognized, as the number of individuals with overweight and obesity continues to grow worldwide.  As the study was 22 weeks in duration, this is a strong element, as most interventions of this nature are much shorter. These results may guide future research study and intervention work to promote exercise and dietary intervention for weight loss and management to improve health outcomes but the lack of detail about the exercise prescriptions provided make this manuscript in need of major revision, as essential information on the exercise programs employed is not currently included.  Additionally, there are several other major and minor concerns that, if addressable, would strengthen this manuscript.

Response: Thank you for your summary and assessment. We also thank the reviewer for your constructive comments that will greatly improve our manuscript. We have tried to clarify your specific comments bellow.

Major Concerns:

Point 1: Title

The reviewer suggests inserting “The effects of” or “No effect of” at the start of the title. Physical activity and exercise are not defined as the same concept (exercise is a subset of physical activity) – as the study measures physical activity levels but also has participants engage in an exercise intervention – the title should choose the term that reflects the study findings most accurately or include both terms as appropriate.

Response 1: The suggestion to insert “the effects of the” was accepted. Regarding the concepts of physical activity and exercise we choose to include both terms. Line 2.

Point 2: Abstract

There is no mention of physical activity assessment or the methods by which they were assessed.

Response 2: This information was added. Lines: 28-29.

Point 3: Introduction

Are the authors able to expand on the duration and intensity elements of exercise programs that are associated with appetite regulation, total caloric intake, macronutrient comp… What is the direction of these relationships and what durations and intensities appear to be optimal?  This may help the reader understand why the Ex Rxs used in the present study were designed as they were. Pg 2; line 62: “Other authors have shown that absolute caloric intake was superior in high intensity exercise.” – Compared to what? Please clarify. Pg 2; line 66: Unclear what a “marker of physical activity level” is?

Response 3: Regarding to the suggestion about expanding the introduction, the findings are still quite inconsistent, but we have included some commentary on that. Lines: 60-61.

In the line 62 (actually 64) the authors found that caloric intake was superior in high intensity exercise compared to moderate intensity exercise. This information was included in the text. Line: 64. Regarding to line 66, we denominate “marker of physical activity level” as daily steps’ categories. We included the explication in the text in order to clarify the reader. Lines: 68-69.

Point 4: Methods

Please expand on the Ex Rx provided to the PA group beyond citing the ACSM Position Stand and include frequency, intensity, modes and duration recommendations for each type of exercise recommended (aerobic, strength & endurance, flexibility). How was the physical activity recommendations delivered to the PA group? Via educational materials, via in person meeting?  Did participants in this group track their activity levels in a measurable way? Additional detail about the S, E, and SE Ex Rx must be included, beyond that they trained 3 days per week. What aspects of physical fitness were targeted?  What modes of training were used?  Were aerobic modes weight bearing or non-weight bearing?  How many minutes were components of the session?  Were sets and reps used for strength training?  What muscle groups were targeted.  What % of 1RM did individuals train at?  How was progression for each component (aerobic, strength) determined?  I see some of this information in Figure 1 – but should be mentioned in text. Please expand on what is meant by “all exercise programs were performed in circuit.” How was adherence to the exercise program measured? What is meant by the “estimated Kcal of the diet?” Is this referring to the recommended Kcal provided by the dietitian? The Motivation Questionnaire should be included as an appendix or additional detail (example questions should be included in the manuscript). Were correlation analyses run by group (rather than controlling for group)? As the data are presented by group in Table 2, it seems as though the correlation analyses were run separately by group, rather than group being a control variable.

Response 4: Additional details about the exercise interventions can be found in in the project methodology article (PMID: 23259716). We chose not to repeat the information in order not to lengthen both, methodology and article. However, if the reviewer considers essential to provide the information again, we can do it.

Physical activity recommendations were transmitted to the participants via a personal meeting. This information was included in the text. Line: 96-97.

Adherence to exercise was calculated by the number of sessions completed in regard to the theoretical sessions ([sessions performed /total sessions]x100) and estimated Kcal of the diet was those calculated and recommended by the dieticians. In order to clarify the reader, this information was included. Lines: 100-102; 124.

Motivation questionnaire questions were included. Lines: 130-140.

Regarding correlation analyses, there was a mistake in the redaction. The reviewer is absolutely right: the analyses were run separately by group, rather than group being a control variable. The error was corrected. Line: 156.

Point 5: Results

Table 1 – what is Physical Activity Level? What is being represented? What are the units? Table 2 - Provide information about control variable as these are presented as partial correlations. Paragraph beginning on pg 4; line 168 – statements need p values or qualifying adjectives to show if changes are statistically significant or not.  

Response 5: In Table 1, physical activity level was the average of the METs during the entire period of use of the accelerometer. It is common for PAL not to be accompanied by units (PMID: 31547205), but we decided to use METs/day as unit. Furthermore, we added more information in the physical activity subsection in Methods section in order to make the text more understandable. Lines: 113-115. On the other hand, in Table 2 there was a mistake and it was corrected. Lines: 156; title of the Table 2. Lastly, p values were included in the cited paragraph and some changes to the text were made to improve understanding. Lines: 193-209. 

Point 6:  Discussion

A paragraph detailing the limitations of the present study should be included.

Response 6: The main limitations were included in the manuscript. Lines: 294-303.
